# In Vitro Antibacterial Activity of Some Plant Essential Oils against Four Different Microbial Strains

Daniela Gheorghita [1], Alina Robu [1,*], Aurora Antoniac [1], Iulian Antoniac [1,2], Lia Mara Ditu [3,4], Anca-Daniela Raiciu [5,6], Justinian Tomescu [6], Elena Grosu [1] and Adriana Saceleanu [7]

1   Faculty of Material Science and Engineering, University Politehnica of Bucharest, 313 Splaiul Independentei Street, District 6, 060042 Bucharest, Romania
2   Academy of Romanian Scientists, 54 Splaiul Independentei Street, District 5, 050094 Bucharest, Romania
3   Microbiology Department, Faculty of Biology, University of Bucharest, Intr. Portocalelor 1-3, 060101 Bucharest, Romania
4   Research Institute of the University of Bucharest, Sos. Panduri 90, 050663 Bucharest, Romania
5   Faculty of Pharmacy, Titu Maiorescu University, 22 Dambovnicului Street, District 4, 040441 Bucharest, Romania
6   S.C. Hofigal Import Export S.A., 2 Intrarea Serelor Street, District 4, 042124 Bucharest, Romania
7   Medicine Faculty, Lucian Blaga University of Sibiu, 10 Victoriei Blvd., 550024 Sibiu, Romania
*   Correspondence: alinarobu2021@gmail.com

**Abstract:** This study evaluates the antimicrobial and antioxidant activities of five essential oils (EO): pine oil, thyme oil, sage oil, fennel oil, and eucalyptus essential oils. To identify the chemical composition of the essential oils, we used gas chromatography coupled to a mass spectrometer (GC-MS). EO are predominantly characterized by the presence of monoterpene hydrocarbons and oxygenated monoterpenes, except in the case of fennel essential oil which contains phenylpropanoids as its main components. The antimicrobial activity of the EO was highlighted on four standard microbial strains (two Gram-negative strains-*Escherichia coli* ATCC 25922 and *Pseudomonas aeruginosa* ATCC 27853; one Gram-positive strain *Staphylococcus aureus* ATCC 25923, and one yeast strain-*Candida albicans* ATCC 10231). Antimicrobial activity was assessed by measuring the diameter of the inhibition zone, and by determining the values of the minimum inhibitory concentration (MIC) and minimum concentration of biofilm eradication (MCBE). Analyzing the diameter values of the inhibition zones we observed increased efficiency of thyme essential oil, which showed the highest values for all tested microbial species. The results of tests performed in a liquid confirm the high sensitivity of the standard strain *Escherichia coli* ATCC 25922 to the action of all essential oils, the lowest values of MIC being recorded for sage and thyme essential oils. For the most essential oils tested in this study, the MCBE values are close to the MIC values, except for the pine EO which seems to have stimulated the adhesion of the yeast strain at concentrations lower than 5%. The study highlights the antimicrobial activity of the tested essential oils on Gram-positive and Gram-negative strains.

**Keywords:** essential oils; antioxidant potential; GS-MS; antimicrobial activity

## 1. Introduction

Essential oils (EO) are produced from plant derivatives and contain between 20 and 60 constituents, the most being part of the terpene family such as hydrocarbons or oxygenated derivatives, esters, and phenols. Research has shown that essential oils have an antimicrobial and antioxidant effect and are often used in alternative medicine [1].

Some essential oils can be used to reduce the pain caused by chronic conditions. For example, peppermint essential oil has analgesic properties and has been shown to reduce pain in patients with osteoarthritis [2]. The antimicrobial activity of essential oils has also been extensively studied in late years and it has been shown that some essential oils can inhibit the growth and multiplication of antibiotic-resistant pathogenic microorganisms.

Due to their antimicrobial activity, essential oils might be considered an alternative to antibiotic treatment. For example, tea tree oil has been shown to have a bactericidal effect on methicillin-resistant *Staphylococcus aureus* microbial strain [3]. Sage, cinnamon, and clove essential oil are often used to relieve respiratory diseases, having an antimicrobial effect on several strains belonging to the species *Escherichia coli, Staphylococcus aureus, Salmonella typhi*, and *Bacillus subtilis* [4]. It is well known that the antioxidant activity of essential oils depends on their chemical composition, namely the presence of phenolic structures and as well as certain ethers, alcohols, ketones, and monoterpenes. These compounds have an important role in some disease prevention by neutralizing free radicals and peroxide decomposition [5,6]. For many years biomaterials are used successfully in various medical specializations such as orthopedic surgery [7–12], dentistry [13,14], general surgery [15–17], and cardiovascular surgery [18,19]. In recent years, herbal medicine has begun to be used to treat many dermatological disorders such as itching and even severe forms of cancer [20,21]. So, the essential oils of *Abies koreana, Anthemis aciphylla, Anthemis nobilis, Citrus aurantium, Eucalyptus globules, Foeniculum vulgare, Mentha* sp., *Salvia* sp. are used in the treatment of dermatological disorders [22–29] such as acne, fungal infection, or cancer. Also, EO from *Afromomum danielli* and *Pogostemon elsholtzioides* reduced blood pressure [30,31], EO from *Salvia officinalis* L., *Citrus aurantifolia, Curcuma longa* L. can improve hyperlipidemia [32–34], while EO from *Lavandula angustifolia* and *Citrus aurantium* reduce blood pressure and anxiety with acute coronary syndrome [35,36]. Peppermint essential oil was used in the acrylic-type bone cement composition, demonstrating its antimicrobial action on *Staphylococcus aureus* and *Pseudomonas aeruginosa* strains [12]. Also, current studies show the effects of using essential oils in bone pain, due to osteoporosis and osteoarthritis [37–44], and in bone repair [45–47]. It evaluated the protective effect of *Rosmarinus officinalis* and *Thymus vulgaris* essential oils against osteoporosis [37], and of the *Ginger, Lavander, Rosmarinus officinalis*, and *Populus alba* essential oils against osteoarthritis [39–41,43,44]. The relevant domains of application of essential oils are shown in Figure 1.

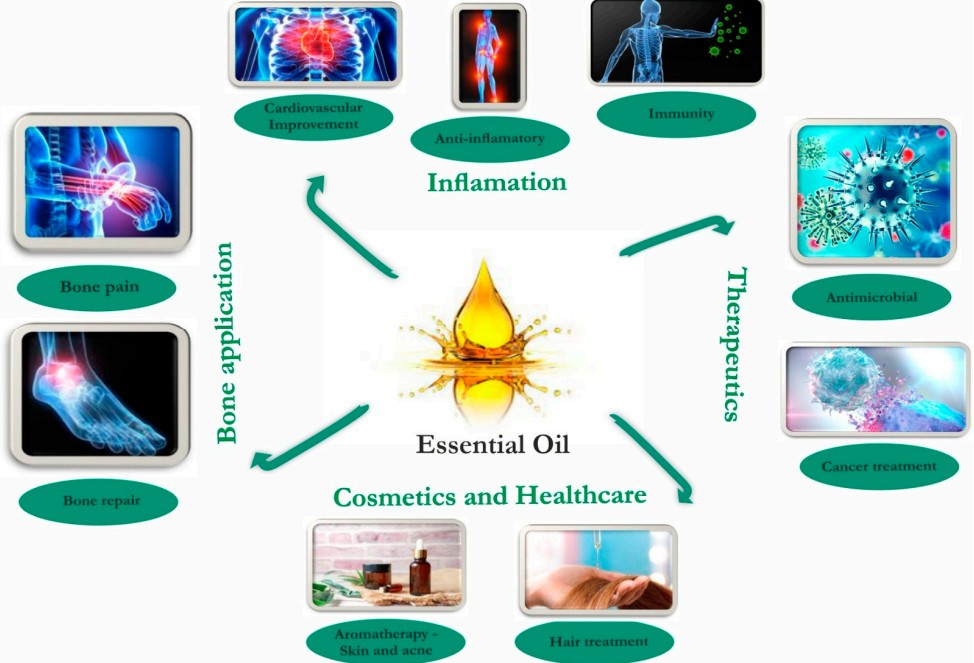

**Figure 1.** Domains of application of essential oils.

**Thyme essential oil** (obtained from the *Thymus vulgaris* plant) has been reported to be widely used to heal burns [48]. The main compound in thyme essential oil, thymol, is active against *Salmonella* and *Staphylococcus bacteria*. It has been shown that the antiseptic properties of thyme help the immune system in chronic infections and is also effective in

chest infections such as whooping cough, bronchitis, and pleurisy [49]. In the study by Dursun et al. [50], the impact of thyme essential oil on burn wounds in rats was investigated, and it was shown that it reduced the amount of nitric oxide produced in response to the burn and facilitated wound healing. The antimicrobial activity of the thyme essential oil was also investigated [51–53]. The study results showed an inhibitory action of this essential oil against *Staphylococcus aureus*, *Streptococci*, and *Salmonella typhimurium*. The major components of thyme essential oil namely carvacrol, p-cymene, γ-terpinene, and α-terpinene have demonstrated antioxidant activity [54–62]. Balahbi A et al. [54] revealed that p-cymene is an important compound used by the pharmaceutical industries for the production of antioxidants. Tepe B et al. [59] compared the antioxidant activity of two thyme essential oils obtained by hydrodistillation with different chemical compositions. The results showed that the antioxidant activity of the investigated oils is due to the presence of a large amount of thymol and carvacrol in their composition.

*Sage* is represented by the whole or cuts leaves of *Salvia officinalis* L. Sage essential oil is rich in thujone [63]. It has a very changeable composition depending on the source, time of harvest, and other factors. The antibacterial properties of sage essential oils are due to the presence of thujone, camphor, and 1,8-cineole in the composition [63,64]. Farhat MB et al. [65] highlighted that the compositions and antioxidant activity of the *Salvia officinalis* L. essential oils present remarkable differences depending on the environmental conditions and geographical origin. In the study conducted by Ozan A et al. [66] on various essential oils from wild and cultivated forms of *Salvia pisidica* it was shown that the antioxidant properties are due to the α-pinene, camphor, and eucalyptol compounds present in the composition of the investigated oils. The results are in accordance with the results reported by Ruberto et al. [67].

*The fennel* plant belongs to the family *Apiaceae (Umbelliferae)*. According to Msaada et al. [68,69], the ripening stages play an important factor in influencing the composition of essential oils, while good agricultural and environmental practices would also help to improve yield and quality. The main constituents of essential oils are: anethole (72.27%~74.18%), fenchone (11.32%~16.35%), and methyl chavicol (3.78%~5.29%), followed by α-pinene, limonene, β-myrcene, camphene, β-pinene, 3-hull, α-phellandrene, cis-anethole, camphor, 1,8-cineole [70]. Fennel oil contains powerful anti-inflammatory compounds, which, when used topically, help with skin care. In a study conducted by Anwar et al. [71], the antioxidant and antimicrobial activities of essential oil, ethanol, and methanol extracts of fennel (*Foeniculum vulgare Mill.*) seeds were examined. The results obtained demonstrate good antioxidant and free radical scavenging activities as well as appreciable antimicrobial activity against selected strains of bacteria and pathogenic fungi for essential oil and various extracts from fennel.

*Eucalyptus* (*Eucalyptus* spp.) is a plant native to Australia grown mainly as a source of fast-growing wood, as well as a source of essential oil used for many purposes. The essential oil is extracted from buds, leaves, bark, and fruits, having an antiseptic, antibacterial, anti-inflammatory, antioxidant, and anticancer action and, therefore, is recommended in the treatment of respiratory diseases like flu, colds, and sinus congestion [72]. The composition of the eucalyptus oil is influenced by geographical location or seasons, which also influences its biological activity. The main compounds of the essential oil are eucalyptol, p-cymene, neo-isoverbenol, limonene, and spathulenol (depending on the species) [73–75]. Salem et al. [76] studied the antioxidant activities of the essential oil from the leaves of *Eucalyptus camaldulensis*, *Eucalyptus camaldulensis* and *Eucalyptus gomphocephala*. It was found that the essential oil of *Eucalyptus gomphocephala* presented the highest antioxidant activity. The study reported that the antioxidant activities of eucalyptus essential oil could be related to phenols such as spathulenol and terpens such as eucalyptol (1,8-cineole).

*Pine* (*Pinaceae*), is one of the most important sources of essential oils in the world, with more than 50 constituents, of which about ten are of key importance [77]. The major components in pine essential oil are pinene, camphene, sabin, carene, myrcene, terpinolene, α-terpineol, limonene, caryophyllene, bornyl acetate, p-cement, felandren, γ-terpene,

germacrene D and spathulenol. Bhalla et al. [78] showed that pine essential oil improves the activity of white blood cells, which are responsible for removing microbes from the body. Terpenoids, the major components of pine essential oil, have proven antimicrobial, antiallergic, antifungal, antiviral, antispasmodic, and anti-inflammatory properties useful in the prevention and treatment of many diseases, including cancer [79–81]. Several types of Pinus are already well-known sources of antioxidants that are commonly used as dietary supplements (against alcohol-induced liver disease [82] or against lipopolysaccharide-induced inflammation, hippocampal memory-enhancing activity, and activity for the early management of dyslipidemia), as phytochemical remedies, and in the treatment of chronic inflammation, circulatory problems [83] and sometimes cancer [84]. Koutsaviti A et al. [83] and Xie Q et al. [85] reported that terpene derivatives such as germacrene α,β-caryophyllene, and γ-terpinene exert antioxidant activity. Also, in the study of Zeng WC et al. [86], the essential oil from pine needles demonstrated significant antioxidant activity, especially against superoxide radicals, and hydroxyl radicals.

The purpose of this paper was to evaluate the physicochemical parameters for five commercial essential oils (fennel, sage, eucalyptus, thyme, and pine essential oils) and their antimicrobial effect on the ability to grow, multiply and generate monospecific biofilms of four standard microbial strains: two Gram-negative strains (*Escherichia coli* ATCC 25922, *Pseudomonas aeruginosa* ATCC 27853), one Gram-positive strain (*Staphylococcus aureus* ATCC 25923), and one yeast strain (*Candida albicans* ATCC 10231).

## 2. Materials and Methods

### 2.1. Obtaining Oils

Five essential oils produced at S.C. Hofigal Export-Import S.A (S.C. Hofigal S.A.) were used in this study. In this process, the post-harvest plant material was sorted and dried in order to prepare it for the extraction of essential oils. The types of EOs investigated, the vegetable material from which the essential oils are extracted, and their origin is presented in Table 1.

**Table 1.** Types of essential oils investigated, plant material used, and its origin.

| Code | Type of EO | The Vegetable Material from Which the Oil Is Extracted | The Origin of the Vegetable Raw Material |
|------|-----------|--------------------------------------------------------|-------------------------------------------|
| FEN | Fennel essential oil (*Foeniculum vulgare*) | Aerial parts | S.C.Hofigal S.A |
| SV | Sage essential oil (*Salvia officinalis*) | Aerial parts | S.C.Hofigal S.A |
| EUC | Eucalyptus essential oil (*Eucalyptus* sp.) | Leaf | Import |
| CI | Thyme essential oil (*Satureja hortensis* L.) | Aerial parts | S.C.Hofigal S.A |
| PIN | Pine essential oil (*Pinus sylvestris*) | Leaf | Spontaneous flora |

The oils are obtained by the hydro-distillation method. In this process, the plant materials are immersed in water and boiled. As a result of the action of temperature on the vegetal matrix, the destruction of the cellular structure occurs, and this leads to the release of aromatic compounds and essential oils. Steam and essential oil vapor are condensed to an aqueous fraction. The vegetable materials used in obtaining the essential oils were subjected to hydro distillation for 6 h using a Clevenger-type device. The advantage of this method is that it protects the oil since the surrounding water forms a barrier to prevent it from overheating. The flow chart for essential oils obtained is presented in Figure 2.

### 2.2. Physical-Chemical Parameters of the Essential Oils

The relative density of the investigated essential oils was determined in accordance with the ISO279:1998 [87], while the refractive index with the protocol recommended by the

European Pharmacopoeia [88]. Relative density at 20 °C was determined using relationship 1 and the refractive index using a Mettler Toledo R40 refractometer.

$$d_{20}^{20} = \frac{m}{m_1} \tag{1}$$

where: $m$ represents the mass of a given volume of oil at 20 °C and $m_1$ is the mass of an equal volume of distilled water at 20 °C

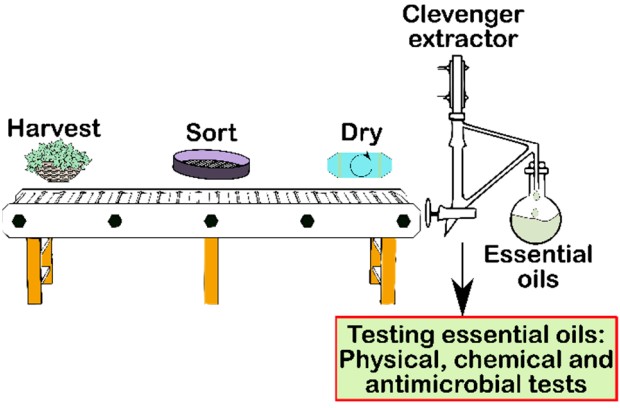

**Figure 2.** Flow chart for essential oils obtaining.

The antioxidant potential of the essential oils has been evaluated by the determination of the ferric reducing antioxidant power (FRAP Assay) described by Benzie and Strain [89]. The reaction underlying the determination is the reduction of ferric Fe(III) to ferrous Fe(II) in the presence of ligand 1.10–phenanthroline at acid pH. A frap unit is defined as the amount of substance needed to reduce one mole of Fe(III) to Fe (II). The calibration curve was obtained using iron sulphate heptahydrate ($FeSO_4 \times 7H_2O$) as standard.

Gas chromatography coupled to a mass spectrometer was used to identify the chemical components in the essential oils. The analysis was performed on a Thermo scientific Trace 1310 gas chromatograph, fitted with a capillary column (TG-WAXMS, 30mx0,25I.D., 0.25 μm film thickness), and equipped with an ISQ 7000 MS detector, from Thermo Scientific, combined with Chromeleon 7 Chromatography data system [90]. Oven temperature was set to 40 °C for 10 min, then ramped at 4 °C/min to 200 °C. After the temperature of 200 °C was reached, it was ramped again at 15 °C/min to 240 °C. The inlet temperature was kept at 250 °C, also the MS detector temperatures for the transfer line and ion source were 250 °C, while the recording mass spectra range was set between 45–250 *m/z*. The samples were injected in split mode, using a 500 mL/min split flow and a split ratio of 333. The carrier gas was helium at a flow of 1.5 mL/min. Sample preparation: 0.1 mL of oil was dissolved in 10 mL of $C_6H_{14}$. The obtained solution was dried from possible existing water droplets by mixing it with anhydrous sodium sulfate powder and then filtered through glass microfiber filters (Whatman CAT No. 1822-070). Compound identification: Peak identification was accomplished by computer software matching using NIST Tandem Mass Spectral Library version 2.4.

### 2.3. Antimicrobial Activities of the Essential Oils

Antimicrobial activities were conducted on four standard microbial strains as is shown in Table 2.

For qualitative testing of antimicrobial activity, 1/10 dilutions in DMSO (Dimethyl sulfoxide) of the five essential oils were performed. To evaluate the ability of microorganisms to adhere to the inert substrate, the purple crystal staining method was performed, and the following solutions were used: acetic acid 33%, purple crystal 1%, and methanol 80%.

**Table 2.** Standard microbial strains used for testing antimicrobial activities.

| No. | Microbial Strain | Source |
|---|---|---|
| 1 | *Staphylococcus aureus* ATCC 25923 | Gram-positive standard strain, The microorganism collection of the Microbiology, Department, Faculty of Biology, University of Bucharest |
| 2 | *Pseudomonas aeruginosa* ATCC 27853 | Gram-negative standard strain, The microorganism collection of the Microbiology, Department, Faculty of Biology, University of Bucharest |
| 3 | *Escherichia coli* ATCC 25922 | Gram-negative standard strain, The microorganism collection of the Microbiology, Department, Faculty of Biology, University of Bucharest |
| 4 | *Candida albicans* ATCC 10231 | Yeast standard strain, The microorganism collection of the Microbiology, Department, Faculty of Biology, University of Bucharest |

*Qualitative evaluation of the antimicrobial effect.* The microbial strains incubated for 18–24 h at 37 °C were used to prepare suspensions with a standard density of 0.5 McFarland ($1.5 \times 10^8$ CFU/mL) (according to CLSI 2021). The suspensions were seeded on agar medium. Subsequently, 10 μL of the essential oils dilutions were distributed in spots. The plates were allowed to stand at room temperature for adsorption of the droplet, after which they were incubated at 37 °C, for 24 h. The antimicrobial efficiency was quantified by the appearance of a growth inhibition zone around the spot. The reading of the results was performed by assessing the clarity of the inhibition zone and measuring and noting the diameter of this zone.

*Quantitative evaluation of the antimicrobial.* For this test, the method of binary serial micro dilutions performed in 96-well plates was used to determine the values of the minimum inhibitory concentration (MIC) represented by the minimum amount of essential oil capable to inhibit the growth and multiplication of the microbial cells. Binary serial dilutions made in a liquid growth medium were subsequently inoculated with 15 mL of standard microbial suspension. After incubation at 37 °C for 24 h, the results obtained by macroscopic observation and spectrophotometric reading at 620 nm, were analyzed.

*Study of the influence of the tested compounds on the development of microbial biofilms on the inert substratum.* The microbial cells were cultured in 96-well plates with a liquid environment and in the presence of different concentrations of test compounds, similar to the method for determining MIC values [91]. After incubation, the plates were washed twice with physiological water. Subsequently, the adhered cells were fixed for 5 min with 150 μL of 80% methanol and were stained with 1% violet crystal alkaline solution (150 μL/well) for 15 min. The staining solution was removed, then the plates were washed under running water. The microbial biofilms formed on the plastic plates were resuspended in 33% acetic acid, and the determination of MCBE values (minimum concentration for biofilms eradication) was performed by spectrophotometric determination of the intensity of the colored suspension, measuring the absorption at 490 nm, using ELISA reader-model SYNERGY HTX multi-mode reader. All tests were performed in triplicate and the results were expressed as an average of the obtained values.

### 3. Results and Discussion

*3.1. Physical-Chemical Parameters of the Essential Oils*

The physical-chemical properties of the investigated essential oils are used as quality parameters according to recommendations established by the European Pharmacopoeia and International Standards. Physical-chemical parameters of the investigated essential oils are presented in Table 3.

The chemical compositions of the essential oils evaluated by GC-MS are presented in Table 4.

**Table 3.** Physical-chemical properties of obtained essential oils.

| Essential Oils | Fennel EO (FEN) | Sage EO (SV) | Eucalyptus EO (EUC) | Thyme EO (CI) | Pine EO (PIN) |
|---|---|---|---|---|---|
| Relative density [g/cm$^3$] | 0.96 | 0.89 | 0.90 | 0.91 | 0.85 |
| Refractive index, λ [nm] | 1.528 | 1.456 | 1.458 | 1.495 | 1.465 |
| Antioxidant activity (mg equivalent to $Fe_2SO_4$x $7H_2O$/g for sample) | 6.09 | 6.54 | 7.09 | 6.79 | 6.49 |

**Table 4.** Main constituents of investigated essential oils.

| Compounds | Classes | Eucalyptus EO | Thyme EO | Pine EO | Sage EO | Fennel EO |
|---|---|---|---|---|---|---|
| β-Pinene | MT | 0.36 | - | 30.21 | 1.77 | - |
| α-Phellandrene | MT | 0.51 | - | - | - | 5.66 |
| β-Phellandrene | MT | - | - | 0.60 | - | - |
| β-Myrcene | MT | 0.26 | 0.62 | 2.81 | 0.66 | - |
| D-Limonene | MT | 8.82 | 0.47 | 18.92 | 1.67 | - |
| Eucalyptol | MT | 82.10 | - | - | 7.27 | - |
| γ-Terpinene | MT | 3.36 | 0.29 | - | 0.39 | - |
| p-Cymene | MT | 4.59 | 35.10 | 3.61 | 0.68 | - |
| Camphene | MT | - | 0.71 | 3.15 | 4.28 | - |
| Pinocarvone | MT | - | 2.08 | - | - | - |
| β-Caryophyllene | ST | - | 2.21 | 3.68 | 3.51 | - |
| α-Caryophyllene | ST | - | - | - | 4.44 | - |
| Carvacrol | MT | - | 58.52 | - | - | - |
| Fenchone | MT | - | - | - | - | 11.91 |
| Estragole | PP | - | - | - | - | 2.97 |
| Anethole | PP | - | - | - | - | 79.46 |
| Carene | MT | - | - | 29.08 | - | - |
| Terpinolene | MT | - | - | 0.59 | 0.25 | - |
| Longifolene | ST | - | - | 2.72 | - | - |
| Bornyl acetate | MT | - | - | 3.05 | - | - |
| α-Terpineol | MT | - | - | 1.59 | - | - |
| Thujone | MT | - | - | - | 38.92 | - |
| Isothujone | MT | - | - | - | 6.19 | - |
| (+)-2-Bornanone | MT | - | - | - | 21.32 | - |
| Linalool | MT | - | - | - | 0.72 | - |
| Borneol acetate | MT | - | - | - | 1.01 | - |
| Terpinen-4-ol | MT | - | - | - | 0.56 | - |
| (-) Borneol | MT | - | - | - | 2.55 | - |
| Viridiflorol | ST | - | - | - | 3.82 | - |
| Monoterpene (MT) | | 100 | 97.79 | 93.6 | 88.23 | 17.57 |
| Sesquiterpene (ST) | | - | 2.21 | 6.4 | 11.77 | - |
| Phenylpropanoids (PP) | | - | - | - | - | 82.43 |

The essential oils of eucalyptus, sage, pine, and thyme are especially characterized by the presence of monoterpenes (monoterpenic hydrocarbons and oxygenated monoterpenes), while the fennel EO contains phenylpropanoids as the main constituents (compounds that can be classified as phenylpropenes or allylbenzenes). Sesquiterpenes are found to a lesser extent in pine, thyme, and sage essential oils. In eucalyptus EO, eucalyptol was identified from the total constituents as the main compound (82.10%), followed by D-limonel (8.82%) and p-cymen (4.59%). From the total constituents identified in the case of thyme EO, carvacrol (58.52%) was the main compound detected followed by p-cymen (35.10%) and other by-products such as pinocarvone, β-caryophyllene. β-pinene, carene, and D-limonene are the main compounds identified in pine EO in a concentration of 30.21%, 29.08%, and 18.92% respectively. In sage EO, 18 compounds were identified, of which thujone and (+)-2-bornanone as majority products in a proportion of 38.92% and 21.32%, respectively. For fennel EO, 4 compounds anethole (majority compound, 79.46%), fenchone, α-phellandrene, and estragole were highlighted.

Thyme EO contains predominantly phenolic terpenoids, such as carvacrol, which explain why exhibited antioxidant activity. The antioxidant activity of eucalyptus EO is due to the presence of the monoterpene compounds 1,8-CineoIe (Eucalyptol) and γ-Terpinene while the antioxidant activity of sage EO is attributed to the presence in the composition of the following compounds 1,8-CineoIe (Eucalyptol), γ-Terpinene, and linalool. β-caryophyllene and terpinolene are compounds that cause the antioxidant activity of pine EO while α-phellandrene and anethole influence the antioxidant activity of fennel EO. The results are sustained by other studies [67,92] that have demonstrated that thymol and carvacrol, are responsible for the antioxidant activity of the essential oils obtained from *Thymus serpyllus* and *Mentha longifolia.* The antioxidant activity of 1,8-CineoIe (Eucalyptol), γ-terpinene, linalool, β-caryophyllene, terpinolene, α-phellandrene, and anethole was demonstrated in many studies carried out on a wide range of essential oils [93–96].

### 3.2. Antimicrobial Activities of the Essential Oils

The qualitative evaluation of the inhibitory effect expressed against different microbial strains was performed by measuring the diameter of the inhibition zone, after incubation of the strains in favorable conditions (Figures 3–6).

Analyzing the diameter values of the inhibition zones (Table 5), it can be observed an increased efficiency of thyme essential oil (CI), which showed the highest values for all tested microbial species, followed by sage and fennel essential oil (FEN), which inhibited the multiplication of species both Gram-positive and Gram-negative bacteria, but did not inhibit the multiplication of yeast species.

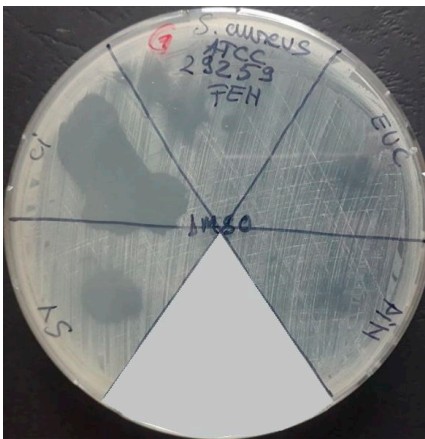

**Figure 3.** The aspect of the inhibition zones expressed by the Gram-positive strain *Staphylococcus aureus* ATCC 25923 in the presence of tested EO at 1/10 dilution in DMSO; FEN (fennel EO); SV (sage EO); EUC (eucalyptus EO); CI (thyme EO); PIN (pine EO).

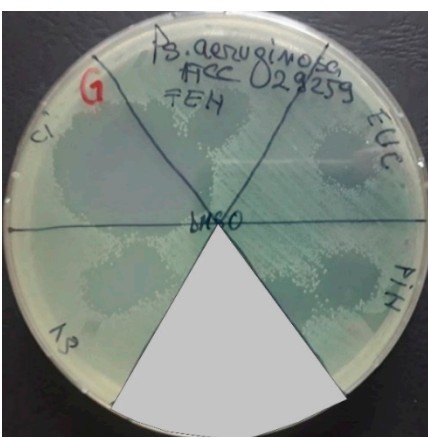

**Figure 4.** The aspect of the inhibition zones expressed by the Gram-negative strains *Pseudomonas aeruginosa* ATCC 27853 in the presence of tested EO at 1/10 dilution in DMSO; FEN (fennel EO); SV (sage EO); EUC (eucalyptus EO); CI (thyme EO); PIN (pine EO).

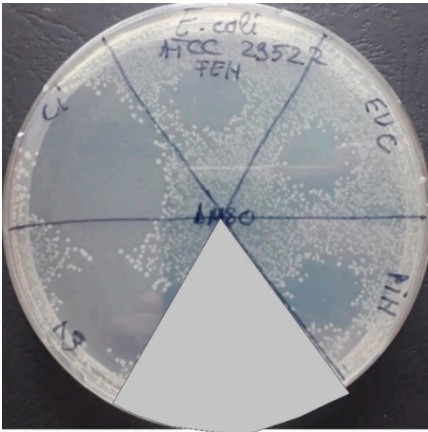

**Figure 5.** The aspect of the inhibition zones expressed by the Gram-negative strains *Escherichia coli* ATCC 25922 in the presence of tested EO at 1/10 dilution in DMSO; FEN (fennel EO); SV (sage EO); EUC (eucalyptus EO); CI (thyme EO); PIN (pine EO).

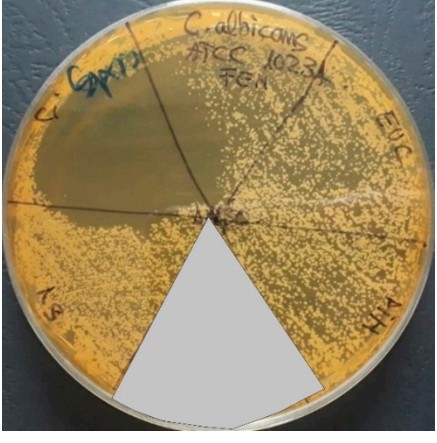

**Figure 6.** The aspect of the inhibition zones expressed by the yeast strain *Candida albicans* ATCC 10231 in the presence of tested EO at 1/10 dilution in DMSO; FEN (fennel EO); SV (sage EO); EUC (eucalyptus EO); CI (thyme EO); PIN (pine EO).

**Table 5.** Inhibition zones of essential oils (dil. 1/10 in DMSO).

| Essential Oils/Microbial Strain | Inhibition Zones [mm] | | | |
|---|---|---|---|---|
| | *Escherichia coli* ATCC 25922 | *Staphylococcus aureus* ATCC 25923 | *Pseudomonas aeruginosa* ATCC 27853 | *Candida albicans* ATCC 10231 |
| **Fennel EO (FEN)** | 16 | 14 | 14 | 0 |
| **Sage EO (SV)** | 22 | 12 | 19 | 0 |
| **Eucalyptus EO (EUC)** | 15 | 6 | 11 | 0 |
| **Thyme EO (CI)** | 30 | 30 | 27 | 35 |
| **Pine EO (PIN)** | 15 | 0 | 12 | 0 |

Although some studies show that essential oils have a more pronounced antimicrobial effect on Gram-positive strains, as opposed to Gram-negative ones, this is not always the case. Some studies show a higher antimicrobial activity of thyme essential oil on *Escherichia coli* strain compared to the Gram-positive strains tested [97].

The most sensitive strain was the standard strain *Escherichia coli* ATCC 25922, which showed sensitivity to all the EO tested, with the largest values of the diameters of the inhibition zones. Also, following a study conducted by Semeniuc and contributors [98], in which the antimicrobial activity of thyme oil on the *Escherichia coli* strain ATCC 25922 was tested by determining the diameters of the inhibition zones, it was observed that the thyme oil has a strong antimicrobial effect on this strain, obtaining a diameter of the inhibition zone of 36–41 mm. Thyme essential oil (CI) has also been shown to inhibit the growth of the *Candida albicans* microbial strain due to its high carvacrol content in its structure, an aspect also highlighted by Tampieri et al. [99].

Other studies in which the antimicrobial activity of thyme essential oil was tested by the spot inoculation method showed that it has a strong antimicrobial effect on the tested strains, obtaining growth inhibition zones with diameters of 26–54 mm. Thyme EO has also been shown to produce a larger growth inhibition zone than those produced by chloramphenicol, suggesting that thyme EO has a stronger antimicrobial effect than chloramphenicol [81].

*Quantitative evaluation of the antimicrobial effect.* The results of tests performed in a liquid medium confirm the high sensitivity of the standard strain *Escherichia coli* ATCC25922 to the action of all essential oils, the lowest values of minimum inhibitory concentration (MIC) being recorded for sage (0.156%) and thyme essential oil (0.626%) as is shown in Table 6 and Figure 7.

The antimicrobial effect of thyme essential oil has been shown in other studies. For example, thyme oil has been shown to inhibit the growth of *Staphylococcus aureus* and *Escherichia coli* strains at concentrations of 0.125% [81].

**Table 6.** MIC values were obtained by the method of serial micro dilutions in a liquid environment.

| Microbial Strain/Essential Oils | MIC Values [% Dilution in Microbial Culture Media] | | | | |
|---|---|---|---|---|---|
| | FEN (Fennel EO) | SV (Sage EO) | EUC (Eucalyptus EO) | CI (Thyme EO) | PIN (Pine EO) |
| *Staphylococcus aureus* ATCC 25923 | 5% | 2.5% | 2.5% | 1.25% | 10% |
| *Pseudomonas aeruginosa* ATCC 27853 | 5% | 1.25% | 2.5% | 1.25% | 10% |
| *Escherichia coli* ATCC 25922 | 5% | 0.156% | 1.25% | 0.625% | 10% |
| *Candida albicans* ATCC 10231 | 10% | 1.25% | 2.5% | 2.5% | 10% |

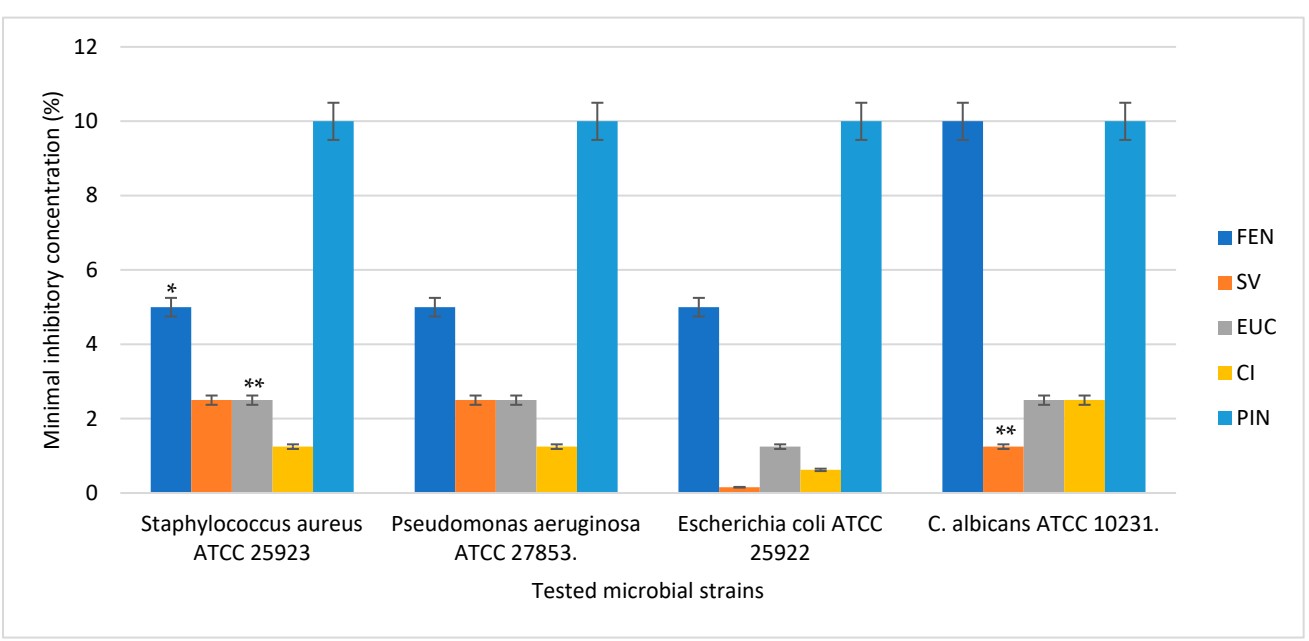

**Figure 7.** Graphical representation of MIC values expressed as a percentage (percentage of oil in the culture medium) for each tested microbial strain; FEN (fennel EO); SV (sage EO); EUC (eucalyptus EO); CI (thyme EO); PIN (pine EO); significant results regarding the MIC values were noted as: * for $p < 0.05$, and ** for $p < 0.01$ ( statistically significant results; statistic interpretation using one-way ANOVA repeated measures test).

Several studies show that many of the essential oils tested have an antibacterial effect on the *Escherichia coli* strain. For example, Xiao and contributors [100] tested the inhibitory effect of 140 commercial essential oils on *Escherichia coli* by determining the MIC values and found that 8 of them, including oregano oil, cloves, and cinnamon, have a stronger antimicrobial effect on *Escherichia coli* strain compared to sufloxacin antibiotic. Also, 40 of the tested essential oils, including cinnamon oil, tea tree oil, and thyme oil, had values of the minimum inhibitory concentration between 0.5–0.125%, and for oregano essential oil, the minimum inhibitory concentration was 0.015% [100].

Contrary to the results of the qualitative tests, the fennel essential oil did not show low MIC values, probably due to its low solubility in the liquid environment, the MIC values being among the highest (5–10%).

*Study of the influence of the tested compounds on the development of microbial biofilms on the inert substratum.* Testing the properties of essential oils to inhibit the adhesion of microbial cells to the inert substrate and to generate monospecific biofilms allowed the establishment of the values of the Minimum Concentration of Biofilm Eradication (MCBE). These values are represented by the highest oil dilutions that inhibited adhesion and are represented in the graphs below (Figures 8–11).

For the most essential oils tested in this study, the MCBE values are close to the MIC values, except the pine EO which seems to have stimulated the adhesion of the yeast strain at concentrations lower than 5% (Figure 10). For *Escherichia coli* ATCC 25922 strain the lowest values of the MCBE were recorded for sage EO (0.04%).

*Staphylococcus aureus* is one of the pathogens that cause most of the recurrent infections associated with biofilm production. In this paper, the lowest values of MCBE for *Staphylococcus aureus* ATCC 25923 were observed in the case of sage and thyme essential oils (SV, CI). Also, a study by Sharifi et al. [101] highlighted the ability of thyme essential oil to eradicate the biofilm produced by *Staphylococcus aureus* isolated from the respiratory tract and from milk samples. This ability of thyme EO to inhibit the adherence of microbial

cells to the inert substrate may be due to the high concentration of thymol, α-terpinol, and carvacrol in its structure.

The ability of sage essential oil to inhibit the adherence of *Escherichia coli* and *Staphylococcus aureus* strains was highlighted in a study by Cui et al. [102]. It was observed that sage oil inhibits their adhesion at concentrations of 0.05–0.1%. On scanning microscopy (SEM) it was observed that sage essential oil destroys the bacterial cell membrane, the ATP concentration decreases from 98.27% to 69.61%, and the nuclear content decreases.

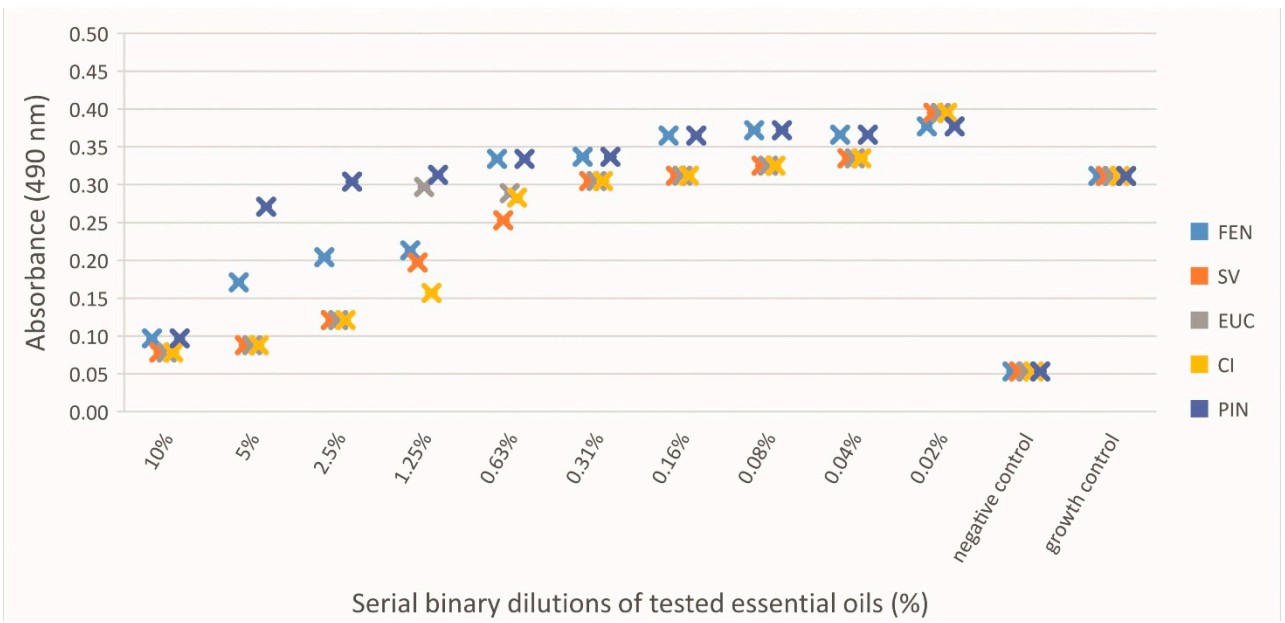

**Figure 8.** Graphical representation of the MCBE values for Gram-positive *Staphylococcus aureus* ATCC25923 strain; FEN (fennel EO); SV (sage EO); EUC (eucalyptus EO); CI (thyme EO); PIN (pine EO).

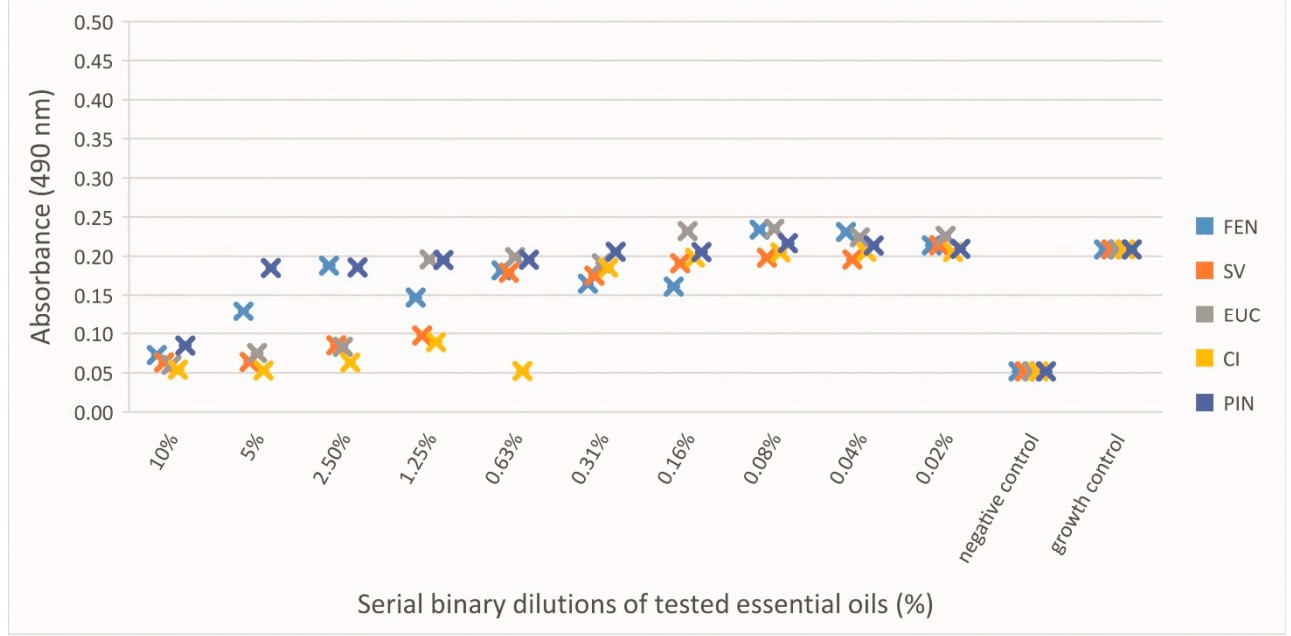

**Figure 9.** Graphical representation of the MCBE values for Gram-negative strain *Pseudomonas aeruginosa* ATCC 27853; FEN (fennel EO); SV (sage EO); EUC (eucalyptus EO); CI (thyme EO); PIN (pine EO).

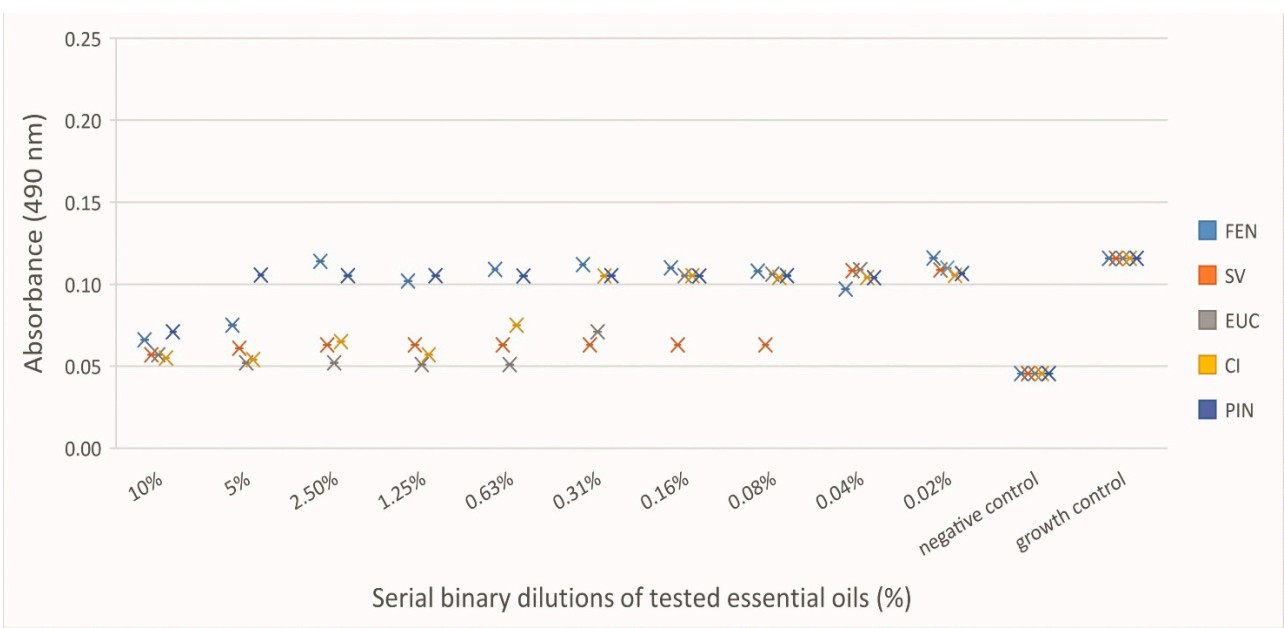

**Figure 10.** Graphical representation of the MCBE values for Gram-negative strain *Escherichia coli* ATCC 25922; FEN (fennel EO); SV (sage EO); EUC (eucalyptus EO); CI (thyme EO); PIN (pine EO).

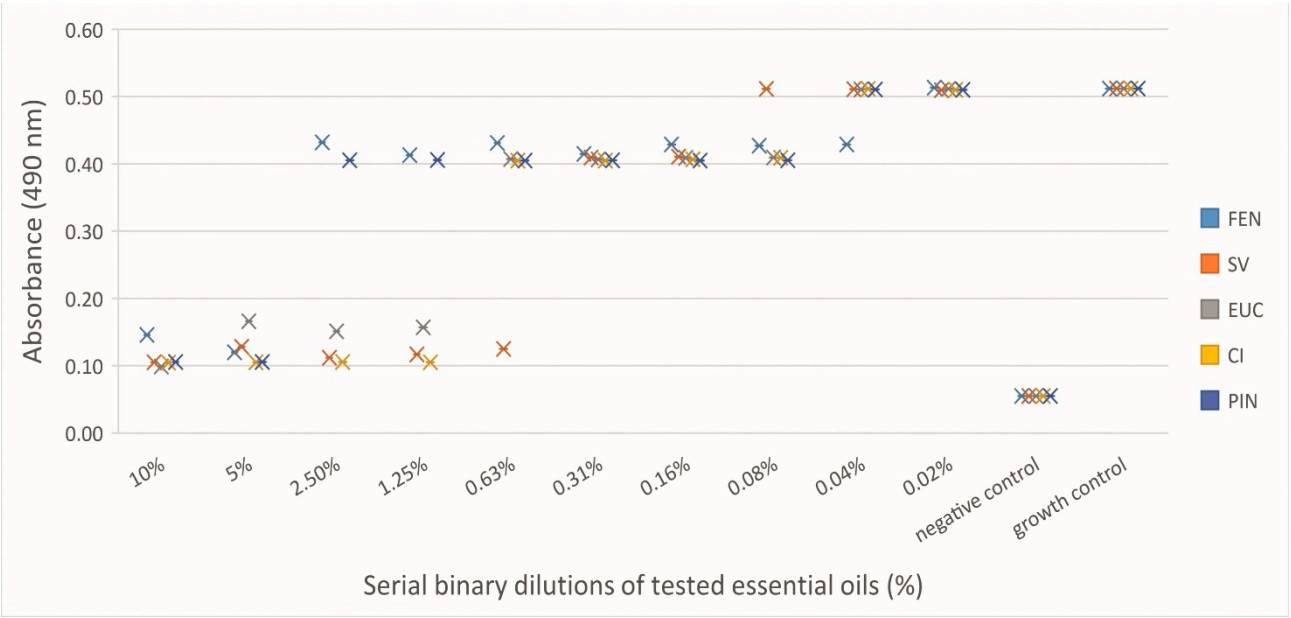

**Figure 11.** Graphical representation of the MCBE values for yeast strain *Candida albicans* ATCC 10231; FEN (fennel EO); SV (sage EO); EUC (eucalyptus EO); CI (thyme EO); PIN (pine EO).

## 4. Conclusions

In conclusion, the results obtained show that essential oils have antioxidant activity, important in the treatment of various inflammatory diseases. The antioxidant activity and composition of essential oils are closely related to the environmental conditions and the geographical origin of the plants from which the essential oils were obtained. The compounds that ensure the antioxidant activity of the investigated essential oils are carvacrol, 1,8-cineole, y-terpinene, linalool, β-caryophyllene, terpinolene, α-phellandrene, and ane-thole. GC-MS analyses revealed that all essential oils are mainly characterized by the presence of monoterpene hydrocarbons and oxygenated monoterpenes, except in the case of fennel essential oil which contains phenylpropanoids as its main components (compounds that can be classified as phenylpropenes or allylbenzenes). This study tested

the antimicrobial activity of five essential oils on the growth, multiplication, and biofilm formation of standard microbial strains: *Staphylococcus aureus* ATCC 2592, *Pseudomonas aeruginosa* ATCC 27853, *Escherichia coli* ATCC 25922, *Candida albicans* ATCC 10231. Both qualitative and quantitative test results show an increased sensitivity of *Escherichia coli* ATCC 25922 strains to all tested essential oils. The qualitative results show the largest diameters of the inhibition zones for thyme EO, followed by sage and fennel EO. Thyme EO was also the only essential oil that expressed an inhibitory effect on *Candida albicans* ATCC 10231 yeast strain. The quantitative results highlighted the lowest inhibitory concentration for sage EO (0.156%) and thyme EO (0.626%). On the opposite, the highest values of the minimum inhibitor concentration were observed for pine EO (10%). The MCBE value was close to the MIC value for most of the tested essential oils.

**Author Contributions:** Conceptualization, D.G., A.A. and I.A.; Methodology, A.R., I.A., L.M.D., A.-D.R. and E.G.; Software, A.R., A.A. and A.S.; Validation, A.A., I.A., L.M.D., A.-D.R., J.T. and E.G.; Investigation, D.G., A.R., A.A., L.M.D., J.T. and A.S.; Resources, A.R., A.A. and E.G.; Data curation, D.G. and A.R.; Writing—original draft preparation, D.G., A.R., A.A. and I.A.; writing—review and editing, D.G. and A.R. All authors have read and agreed to the published version of the manuscript.

**Funding:** This work was supported by a grant from the Romanian Ministry of Education and Research, CCCDI-UEFISCDI, Project number PN-III-P2-2.1.-PED-2019-5236, within PNCDI III. In addition, financial support from the Competitiveness Operational Program 2014-2020, Action 1.1.3: Creating synergies with RDI actions of the EU's HORIZON 2020 framework program and other international RDI programs, MySMIS Code 108792, Acronym project "UPB4H", financed by contract: 250/11.05.2020 is gratefully acknowledged.

**Institutional Review Board Statement:** Not applicable.

**Informed Consent Statement:** Not applicable.

**Data Availability Statement:** The experimental data on the results reported in this manuscript are available upon on official request to the corresponding authors.

**Conflicts of Interest:** The authors declare no conflict of interest.

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
