# Peer review of "In Vitro Antibacterial Activity of Some Plant Essential Oils against Four Different Microbial Strains"

_applsci, doi:10.3390/app12199482_

Round 1
Reviewer 1 Report
Revision is attached in a separate document.

Author Response
Dear reviewer,
The Authors of the manuscript entitled “In vitro antibacterial activity of some plant essential oils against four different microbial strains” submitted to APPLIED SCIENCES thank the reviewer for reviewing our manuscript. We are deeply grateful for the observations and comments that we addressed and feel that greatly increased the quality of our manuscript. Please find in the attachment the answers to all comments and suggestions.

Reviewer 2 Report
The authors of this study are asking an urgent question, namely, they are trying to find new antimicrobial and antioxidant substances that can be isolated from oils. Gheorghita and colleagues evaluated the antimicrobial and antioxidant activity of five essential oils (EO): pine oil, thyme oil, sage oil, fennel oil and eucalyptus essential oils. Antimicrobial activity was assessed by measuring the diameter of the inhibition zone and determining the values of the minimum inhibitory concentration (MIC). The results of tests carried out in liquid confirm the high sensitivity of the standard strain of the bacterium Escherichia coli ATCC 25922 to the action of all essential oils, the lowest values were recorded only for essential oils of sage and thyme. The text is written in simple and accessible language, the narration of the chapters is consistent. However, there are a number of significant comments that should be clarified in the work. After correcting these comments, the work can be published in the journal Applied Sciences.
(1) Table 3. What are the reasons for the excellent Refractive Index and Antioxidant activity? The density values for all compounds are within the margin of error.
(2) Strengthen the introduction of antioxidant activity. A very extensive block for antimicrobial, unequal. This also applies to the conclusions of the work.
(3) Insert links to several defining works. a) Prabuseenivasan S., Jayakumar M. and Ignasimuthu S. Antibacterial activity of some essential oils of plants In vitro. BMC Supplements Alter Med 6, 39 (2006). https://doi.org/10.1186/1472-6882-6-39 . b) Abdel-Fattah, S. M., Abo-Sria, Y. H., Abu-Seif, F. A. and Shaaban, H. A. (2008). Antibacterial activity of some essential oils of plants In vitro. Journal of Food and Dairy Sciences, 33(12), 8577-8590.
Author Response
Dear reviewer,
The Authors of the manuscript entitled “In vitro antibacterial activity of some plant essential oils against four different microbial strains” submitted to APPLIED SCIENCES thank the reviewer for reviewing our manuscript. We are deeply grateful for the observations and comments that we addressed and feel that greatly increased the quality of our manuscript. Please find in the attachments the answers to all comments and suggestions.

Round 2
Reviewer 1 Report
Dear Authors
I found that manuscript was substantially revised also I appreciate responses on all my comments.